# Mineral Nutritional Status of Yaks (*Bos Grunniens*) Grazing on the Qinghai-Tibetan Plateau

**DOI:** 10.3390/ani9070468

**Published:** 2019-07-23

**Authors:** Qingshan Fan, Metha Wanapat, Fujiang Hou

**Affiliations:** 1State Key Laboratory of Grassland Agro-ecosystems; Key Laboratory of Grassland Livestock Industry Innovation, Ministry of Agriculture; College of Pastoral Agriculture Science and Technology, Lanzhou University, Lanzhou 730020, China; 2Tropical Feed Resources Research and Development Center (TROFREC), Department of Animal Science, Faculty of Agriculture, Khon Kaen University, Khon Kaen 40002, Thailand

**Keywords:** macro and micro minerals, correlations, alpine meadow, yak, Qinghai-Tibetan Plateau, grazing

## Abstract

**Simple Summary:**

The investigation was carried out to study the mineral status of yaks (*Bos grunniens*) grazing on the Qinghai-Tibetan plateau (QTP) in four counties. Samples of soils, forages, and serum of yaks were collected and analyzed for both macro and micro mineral concentrations. The results revealed the deficiency of both macro and micro minerals in both seasons. The data are useful for further research, as well as the possible supplementation of both macro and micro minerals, to improve the health and productivity of the yaks in the QTP region.

**Abstract:**

Minerals are essentially important for supporting livestock’s health, as well as productivity. This study aimed to investigate the mineral status of yaks (*Bos grunniens*) grazing on the Qinghai-Tibetan Plateau (QTP) and the relationship between macro and micro mineral nutrients among soil, forages, and blood in four counties of the QTP. The soil samples (*n* = 320), forages (*n* = 320), and blood serum (*n* = 320) were collected from four randomly selected yak farms in each location during July (warm season) and December (cold season), and were analyzed for macro minerals (P, Ca, K, Mg, Na) and micro minerals (Fe, Mn, Zn, Cu, Se). Based on this study, both of the macro and micro minerals were very variable between seasons and many of the macro and micro minerals, such as P, Mg, K, S, Na, Se, and Cu, were found to be below the requirement level for yaks in all four counties. It was significantly shown that the concentrations of both macro and micro minerals in soil and forages influenced the serum concentration of minerals, showing the deficient status of yaks.

## 1. Introduction

Livestock grazing plays an important role in the element cycling of rangeland ecosystems, including the alpine meadow of the Qinghai-Tibetan plateau (QTP) [1]. The energy and protein contents of herbage are generally viewed as the most important factors influencing livestock production [2]. However, an ever increasing amount of evidence indicates that minerals govern the health of large herbivores [3]. Minerals are essential for animals to maintain osmotic pressure and an acid-base balance, as catalysts in enzyme and endocrine systems, and in the regulation of cell replication and differentiation [4]. Numerous studies have shown that macro and micro mineral nutrient deficiencies can lead to impaired growth and reproduction and increases in disease, due to impaired immune function, and serious deficiencies may even cause death [5]. Mineral requirements of animals depend on many factors, namely the production system and the environment [6]. In many prevailing areas of the world, however, pasture, which constitutes most ruminant foods, cannot fully support the mineral requirements of grazing ruminants all year round [7,8]. Soil-borne mineral nutrients are taken up by yaks in the herbage and shrubs of the alpine meadow, and also at times in a small amount, consumed directly from the soil (Figure 1) [9,10]. The content of mineral elements in herbage has a crucial influence on the content and the balance of mineral elements in livestock, and indeed a mineral deficiency in herbage will predispose to a deficiency in the blood serum concentration of grazing livestock [11]. For this reason, the contribution of soil type and resulting herbage nutrient composition greatly contribute to livestock performance [12]. In grazing grassland, the content of mineral elements in soil and herbage will eventually be reflected in livestock blood serum and this productivity [13].

The yak (*Bos grunniens*) is an important domesticated ruminant inhabiting the QTP in China [1]. Today, there are approximately 14 million yaks raised on the plateau, representing about 95% of the world’s yak population. These ruminants are vital for the livelihoods of the herders, providing them with milk and meat for consumption, fiber for clothing, and dung for fuel [14]. The traditional farming systems suffer from seasonal fluctuations in forage availability, with severe shortages of herbage during the long cold season [1]. Under the grazing system, pasture herbage cannot sufficiently provide the mineral requirements for grazing yaks. Whilst there have been a number of studies and programs aimed at improving yak production through mineral supplementation [15], there is a relative paucity of data on mineral nutrition in livestock compared to our knowledge of protein and energy requirements [6]. An assessment of minerals contained in soil and herbage where livestock graze is considered an important protocol [13]; however, the mineral profile of soil, plants, and animals has so far not been studied in detail for the QTP [8]. The objective of this study is to better evaluate the nutrient profile of mineral elements in grazing livestock by analyzing the content and correlation of mineral elements in soil, grass, and livestock in this area, and to provide a scientific basis for the supplementation of mineral elements and the improvement of livestock productivity.

## 2. Materials and Methods

### 2.1. Study Site and Vegetation

The study areas were selected from four country sites located in the rangeland grazing areas of the mid-eastern part of the QTP, as follows: Datong County in Qinghai Province (latitude 37°13′24″N, longitude 101°21′36″E, elevation 3100 m); Maqu County in Gannan Tibetan Autonomous Prefecture, Gansu Province (latitude 33°40′4″N, longitude 101°52′12″E, elevation 3504 m); Hongyuan County, Aba Prefecture, Sichuan province (latitude 33°40′54″N, longitude 103°52′27″E, elevation 3900 m); and Naqu County, Tibet Autonomous Region (latitude 30°58′68″N, longitude 91°37′34″E, elevation 4700 m) (Figure 2). Climate data for the sites are summarized in Table 1. The grazing sites were randomly selected at the study sites on the QTP.

The various rangelands of the plateau are characterized by their high altitude, very low annual average temperature (from −1.5 to 1.2 °C), short growing season (from June to September), and great seasonal variation in feed supply [15]. Soils of the study sites are classified as alpine meadow soil.

### 2.2. Animals Management

The animal sampling procedure strictly followed the rules and regulations of experimental field management protocols (file No: 2010-1 and 2010-2), which were approved by Lanzhou University.

Forty yaks with an average initial body weight of 258.26 ± 9.35 kg aged 4 years, were randomly selected for each county. All yaks grazed on herbage following the recommendations of the common raising practice, which involves animals remaining permanently at pasture during the warm season (June to September), but being penned at night during the cold season (October to May).

### 2.3. Soil, Herbage, and Blood Sampling

Samples of soil, herbage, and blood serum were collected from four randomly selected yak farms of each district. From each sample farm, 10 soil and 10 herbage samples were randomly collected following two “W” shapes across approximately 400 to 500 ha of alpine meadow pasture, during July (warm season) and again in December (cold season) of 2018. Soil samples were taken at a 15 cm depth; in total, 320 soil samples (warm and cold season) were collected from all four districts. After sun-drying, soil samples were processed through a 0.25 mm sieve for laboratory analysis. Herbage samples were collected by cutting off the top portion, dried at 105 °C for 12 h, and bulked for chemical analysis to produce overall means for both the warm and cold season. In total, 320 herbage samples (warm and cold season) were collected from all four districts. Blood samples (10 mL) from the jugular vein were drawn from each yak. In total, 320 blood samples were taken in July (warm season) and again in December (cold season) from 10 adult yaks per farm. Each blood sample was collected and centrifuged for 15 min at 2000 g, and the supernatant serum was then collected in polyethylene tubes and stored at −20 °C until analysis.

### 2.4. Mineral Analysis

Approximately 0.2 g of each of the dried soil sample was digested for 20 min at 140 °C and 15 atm in 5 mL of concentrated nitric acid (‘suprapur’ grade), 2 mL hydrochloric acid, 1 mL hydrofluoric acid, and 1 mL of 30% *w/v* hydrogen peroxide. The digested samples were cooled to room temperature and transferred to a Teflon cup, 1 mL perchloric acid was added, and the hydrofluoric acid was removed for 10 min at 180 °C, and then used for the analysis of total selenium in the soil [10].

Approximately 0.2 g of each of the dried herbage samples was digested for 5 min at 140 °C and 15 atm in 5 mL of concentrated nitric acid (‘suprapur’ grade) and 1 mL of 30% *w/v* hydrogen peroxide. A total of 200 μL of each serum sample was digested for 4 min at 140 °C, at 14 atm in 5 mL of concentrated nitric acid (‘suprapur’ grade) in a microwave digestion system (WX-4000, Shanghai Qiyao Ltd. Co). The digested samples were transferred to polypropylene sample tubes and diluted to 100 mL with ‘ultrapure’ water (Sartorius Arium 611 DI) [10].

Samples (soil, herbage, and serum) were analyzed for Ca, Mg, K, Na, P, Fe, Zn, Mn, and Cu by the ICP-AES analyzer (IRIS Advantage ER/S) [16], and the analysis of Se was carried out by atomic fluorescence spectrophotometry [17].

### 2.5. Statistical Analysis

The soil, herbage, and yak blood serum concentrations, including [mineral element]_plant_/[mineral element]_soil_, [mineral element]_yak_/[mineral element]_plant_, and [mineral element]_yak_/[mineral element]_soil_, were statistically determined as a split-plot [18], with the seasonal class as the main plot and region as the subplot. Significance levels used were 0.05 to 0.001. Differences between means for soil, herbage, and yak blood serum concentrations, including [mineral element]_plant_/[mineral element]_soil_, [mineral element]_yak_/[mineral element]_plant_, and [mineral element]_yak_/[mineral element]_soil_, were ranked using Duncan’s New Multiple Range Test.

The General Linear Model was used for ANOVA testing of mineral concentrations in soil, herbage, and blood serum for different districts. Correlation coefficients of the mineral content in soil, herbage, and yak were determined from the data for mineral levels of soil, herbage, and blood serum, and the correlation between the assessed elements was estimated by Pearson’s product-moment correlation coefficient. The regression equations on the relationship between the soil–plant, plant–animal, soil–animal, and soil–plant–animal were determined using a linear regression model. All statistical calculi were performed using software SPSS version 17.0 of Statistical Software Package (SPSS Inc. Chicago, USA). A plant’s ability to take up elements from the soil, yak’s ability to take up elements from the plant, and yak’s ability to take up elements from the soil were evaluated by the Translocation Factor (TF), expressed by the following ratios: [mineral element]_plant_/[mineral element]_soil_, [mineral element]_yak_/[mineral element]_plant_, and [mineral element]_yak_/[mineral element]_soil_.

## 3. Results

The effective contents of most elements in soil were affected by regional differences (*p* < 0.05), and some elements, such as Ca, K, P, Na, Fe, Zn, and Cu, were also significantly affected by the season (*p* < 0.05) (Table 2). Based on the recommended levels reported for soils, it was observed that the soils of the study area were classified as a high recommended level for Ca, K, Fe, Mn, Zn, and Cu, but the concentrations of P, Mg, and Se in soil were extremely below the range of the recommended requirement for both seasons of summer and winter in all four regions. When comparing the results for the summer and winter in each county, only the K, Fe, Mn, Zn, and Cu contents in the soil in the summer indicated a significantly higher level than winter (*p* < 0.05). A relatively higher concentration of soil Mg and Ca occurred in summer than winter (*p* < 0.05), with the exception of Hongyuan County.

Except for Mn, macro and micro minerals in herbage were significantly affected by season (*p* < 0.05), and the content of K, P, Mg, Na, Zn, and Cu in herbage in all regions was significantly higher in summer than in winter. In addition, some elements were affected by regional differences (*p* < 0.05), and the concentrations of K, Na, Zn, Cu, and Se were also significantly affected by the interaction between regions and seasons (*p* < 0.05) (Table 3). The Mn concentration was relatively low in winter compared to summer (*p* < 0.05), except for Naqu County. Most of the herbage samples were deficient in Se all year round. The herbage Na concentration was in an exceptionally low range of requirements for yak recommended (Table 3) for all regions all year round. K, P, and Cu contents in herbage of different regions were invariably higher than the recommended level in summer, but lower than the recommended level in winter. The content of Fe in herbage was much higher than that of the recommended level.

The concentrations of Ca, K, P, Mg, Fe, Mn (except for Hongyuan County (HC)), Zn, and Cu in the blood serum of yak were influenced by the season (*p* < 0.05), and were significantly higher in the summer than in the winter (*p* < 0.05) in all counties (Figure 3). In addition, all the mineral elements were affected by regional differences (*p* < 0.05). Serum concentrations of Na were relatively lower than the marginal range for all values during the summer and winter (except for Maqu County (MC)). The serum Na concentration was variable among counties and was significantly different between the winter and summer, especially in the MC and HC counties. The concentrations of P and Cu in the blood serum of yak were approximately two times higher in the summer when compared with the recommended level in all regions. However, the blood serum contained relatively lower P and Cu levels in the winter, and these were lower than the minimum values recommended for yak. Our results revealed that most of the yaks were deficient in Se and Na during both the summer and winter.

The transfer factor (TF) was calculated for those elements detected in the soil, herbage, and serum of yak (Table 4, Table 5 and Table 6). On average, mobility was higher within the soil than in herbage to yak, except for Se during the summer and winter. Herbage bioaccumulation from the soil showed TF > 1 for all elements except Se during the summer and winter. In particular, P and Se showed the highest and lowest values of the herbage/soil ratio in summer, respectively; however, Mg and Se showed the highest and lowest values of the herbage/soil ratio in winter. Na and Fe showed the highest and lowest values of the yak/herbage ratio during summer and winter, respectively. Na and Fe showed the highest and lowest values of the yak/soil ratio during summer and winter, respectively. Among the elements detected, the highest translocation between soil and herbage occurred in P, Mg, K, Na, and Ca during summer and winter; between soil and herbage in Na during summer and winter; and between herbage and yak in Na during summer and winter. In particular, Na was by far the most mobile element between soil and yak and herbage and yak during the summer and winter.

Significant correlation values were obtained between soil and herbage for K, Mg, Mn, Cu, and Se in summer. Significant correlation values were obtained between soil and herbage for P, Mn, Zn, and Cu in winter. The correlation values between herbage and yak were significant for all the minerals studied except for P in summer. Significant correlation values were obtained between soil and herbage for Fe, Zn, Cu, and Se in winter. No significant correlation values were obtained between soil and yak for Ca, P, Na, and Fe in summer. Significant correlation values were obtained between soil and herbage for Ca, K, Mg, Cu, and Se in winter (Table 7).

## 4. Discussion

The herbage Na content in this study was found to be extremely low, regardless of seasons, and only met more than about one tenth of the recommended dietary requirements (800–1200 mg/kg), as stated by Freer et al. [20], which could explain, at least in part, the low concentrations of Na in yak serum. However, the extent of the deficiency of serum Na of yak was not too serious, even though the levels were below the minimum level recommended by McDowell [5,21]. Natural herbage and soil were the only sources of minerals for the yaks in this study, as they did not receive any supplementation and were not provided with mineral licks. Xin et al. [6] stated that the relative sufficient Na level in yak serum could be due to soil digestion by licking soil in the winter, and the amount of soil consumed during the winter would be sufficient as dried and wilted portions of herbages are much smaller, even in bare land. These would partly be explained by the altitude of deficiency in Na from herbage and also by the available higher serum Na of yak in the winter. The current study indicated that [mineral element]_yak_/[mineral element]_soil_ was 20 times higher than [mineral element]_yak_/[mineral element]_plant_. This would partially explain why the extent of deficiency in Na requirement from forage and serum samples were so significantly different in the same season. This may have occurred in yak during the summer in this study as the animals were rapidly recovering from their loss of body weight in previous winter periods, and could explain the relatively higher serum Na level in winter. These phenomena, therefore, lead to Na deficiency becoming worse in summer than winter.

Grasses contain a relatively higher concentration of P during early growth, but decline rapidly as herbages mature and senesce [22]. Herbage had sufficient P in summer, but was deficient during winter in the northwest region of China. The current study indicated that [mineral element]_plant_/[mineral element]_soil_ during the summer was three times higher than in winter. This would partly explain the relatively higher herbage P in summer. All the mean concentrations of serum P were below the recommended level of 46.5 mg/L [5] during winter. This result showed that the risk of P deficiency appears to be widespread in yak during winter. The results are consistent with the findings reported by Long et al. [23], in which a deficiency of P was a recognized problem in grazing yaks in late winter. The lower dry matter intake for grazing livestock with a P deficiency can result in a reduction in fiber digestion by limiting microbial activity, and reduce microbial protein synthesis, thus reducing the intestinal absorption of amino acids (AA) so that animals become AA deficient [5].

Earlier studies have shown that the concentrations of K in herbages decline with grass maturity [5]. Our results presented that K concentrations in herbage were five times higher in summer than winter, and the K concentrations in winter forages were below the recommended level of 5.0 g/Kg [20] in all counties. This was consistent with a former report on K concentrations in forages from the northwest of China [24]. In the present study, although concentrations of K in soils were high in terms of the recommended level during the summer and winter yearly, the concentration of mineral in herbage obtained was lower than the recommended level in winter, as [mineral element]_plant_/[mineral element]_soil_ during the summer was three times higher than in winter. However, serum K concentrations were found to be at the marginal level of 97.5–234.0 mg/L in all study areas [5]. The high yak serum of K concentration could be attributed to the soil licking of yak when grazing on the herbage during the winter season, as [mineral element]_yak_/[mineral element]_soil_ was 33 times higher than [mineral element]_yak_/[mineral element]_plant_.

In this study, the Ca concentration in the soil was about five times higher than the recommended level of 72 mg/kg DM [19] during the summer and winter. Higher levels of Ca contained in soil may increase Ca concentrations in herbage [25]. This is consistent with the current research results, for which all the concentrations of herbage Ca were within the recommended range of 2–11 mg/kg DM [20]. Furthermore, serum Ca levels were above the recommended level (90 mg/L) in all regions and for both seasons. These results were similar to those reported earlier by Masters [24]. Nevertheless, the serum Ca concentrations of yaks were higher in the summer than in the winter, in agreement with the results of Zhou et al. [26]. This may be due to a lower vitamin D status in yak in winter, and the ability of animals to absorb and use Ca depends on the supply of vitamin D [22].

Ashraf et al. [27] reported that although some mineral elements in the soil are rich, some element deficiency may still occur in the herbage. This is contrary to the current research results, which show that, although the Mg concentration in soils was lower than the recommended level of 30.0 mg/kg [19] during the summer and winter, the concentrations of herbage minerals ranged well within the requirement level [20]. This finding is similar to that illustrated by Kumaresan [8]. Accordingly, serum Mg concentrations (28.4–42.5 mg/L in serum) in yaks were all above the marginal range of 25.0 mg/L [5]. Therefore, the yaks had a good status for Mg sufficiency all year round. The results were in agreement with earlier values which demonstrated that Mg deficiency for sheep seldom occurred in northwest China [24].

The present investigation showed that there were significant seasonal effects on the concentrations of Fe, Zn, and Cu in the soil, particularly in Fe and Cu. The contents of Fe, Zn, and Cu in soil were higher in summer than in winter, which may be related to the soil temperature and microbial activity. The average concentrations of trace element values of soils in China were as follows: [19]: Fe (2.5 mg/kg), Mn (5 mg/kg), Zn (2.5 mg/kg), Cu (0.3 mg/kg), and Se (0.5 mg/kg). The results of the present study revealed that the surface soils were much higher compared with the average value of soils, except for Se during summer and winter. Se deficiency results in white muscle disease, an illness that causes high mortality in young calves and lambs. White muscle disease is not a serious problem in areas where the soil is high in Se [28]. However, the current study suggested that soil Se content is extremely deficient. Fe contents were relatively high. High concentrations of Fe in the soil are often related to Fe toxicity [29]. The present study shows that the effective content of Fe in soil is nearly 100 times higher than the recommended standard.

The present experiment revealed that there were significant seasonal effects on the concentrations of micro minerals in the herbage, particularly in Fe, Zn, Se, and Cu. The seasonal variations in minerals in forage are mainly attributed to the stage of maturity of forage [30]. As plants mature, mineral contents decline due to a natural dilution process and the translocation of nutrients to the root system [21]. This is consistent with the current research results, in which the Zn and Cu concentrations of forages were higher in summer than in winter. However, the content of Fe and Se in herbage was higher in winter than in summer, which might be related to soil pollution in herbage. Herbage varies widely in micro mineral content due to soil type, pH, vegetation type, and horizontal distribution [31]. In the present experiment, it was evidenced that Zn, Fe, and Mn concentrations in herbage samples were higher than the recommended level [20]. The insufficiency of race mineral concentrations in the forages will finally lead to a deficiency in serum concentrations of grazing animals [21], and this occurred in the present study for marginal deficiency in Cu (cold season) and deficiency in Se (both summer and cold season). The high content of available Fe in the soil contributes to the absorption of elemental Fe by plants [8], which may be responsible for the high levels of herbage Fe.

The abnormal content of mineral elements in animals, especially in the blood, kidney, liver, and other parts, can reflect the animal’s body in a certain disease or poisoning state [32]. Zn and Cu are the most important essential trace elements, playing a significant role in the growth and development of animals [33]. In the ruminants, average blood Cu values of <0.5 μg/mL are a sign of severe Cu deficiency [26]. Cu deficiency has been associated with the abnormal growth of fur, impaired growth and reproductive performance, heart failure, and gastrointestinal disturbances [28]. The mean concentration of Cu observed in the present study was lower than the recommended value in winter. The data are similar to the results given by Moscuzza et al. [34], in which the risk of Cu deficiency is a common occurrence in bovine production. Underwood and Suttle [22] showed that serum Fe, Mn, Zn, and Se in yak should be 1.1 mg/L, 0.006 mg/L, 0.6 mg/L, and 0.003 mg/L, respectively. These levels of serum Fe, Mn, and Zn are remarkably higher than the recommended value; however, the content of elemental Se in the blood of yak is below the recommended value during summer and winter. Se deficiency interferes with the normal growth processes of sheep and cattle. Se deficiency also disrupts the normal reproductive process, apparently affecting ovulation and fertilization, resulting in a higher incidence of embryonic mortality [28]. Masters et al. [35] pointed out that when the total Se content of the soil is less than 0.5 mg/kg DM, a lack of elemental Se can occur in livestock grazing in the area. The soil Se content in this study was found to be less than 0.5 mg/kg DM (Table 1), which further explains the lack of Se in the region.

K, Mg, Mn, Cu, and Se showed significant correlation coefficients in soil and herbage in summer and P, Mn, Zn, and Cu in winter, suggesting the potential use of herbages as a bioindicator of macro and micro minerals. The mineral contents of the herbage depend upon the type of soil and environmental conditions in which they are grown [31]. However, the effectiveness of the element is often reduced by the influence of soil properties, including particle size, pH, and water content, resulting in a decrease in the effective content of mineral elements in the soil [36]. As reported, there was a close relationship between soil minerals and herbage mineral contents, and if in the level of essential minerals is low, the uptake by roots will be impaired [25]. The correlation coefficient values between herbage and yak serum were significant for Ca, K, Mg, Na, Fe, Mn, Zn, and Se, and Fe, Zn, Cu, and Se, respectively. However, such correlation coefficient values were not found between the mineral levels in yaks and mineral levels in soil, except for K, Mg, Mn, Zn, Cu, and Se in summer, and Ca, K, Mg, Cu, and Se. These values are consistent with the findings of Kumaresan et al. [8], who reported that correlations were found between soil, herbage, and the blood of cattle in a subtropical hill agro−ecosystem. In the present study, the regression equation developed to predict the mineral concentration in yak based on the soil and herbage mineral content showed a positive relationship for K, Mg, Mn, Zn, Cu, and Se in summer, and Ca, K, Mg, Mn, Zn, Cu, and Se in winter, suggesting the mineral status in yaks.

## 5. Conclusions

The present study confirms the prevalent deficiency of Na and Se and seasonal deficiency of P and Cu in the serum of yaks and could be the predisposing factor to livestock grazing under traditional conditions. Further studies with the mineral supplementation of yaks should be conducted in order to identify the effect of individual mineral and multi−mineral combinations on grazing yak production on the QTP. Furthermore, the strong positive correlations between macro and micro mineral concentrations in herbage and yak serum suggest that herbage is potentially useful for monitoring the health of yaks in general, and most elements examined, particularly in the warm season; however, the strong positive correlations between micro and macro mineral concentrations in the soil and yaks suggest that the soil is potentially useful for monitoring the mineral status and health of the yaks grazing, especially in the winter.

## Figures and Tables

**Figure 1 animals-09-00468-f001:**
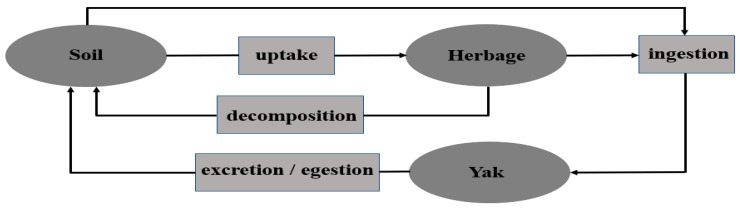
The cyclic path of mineral elements in a grassland grazing system.

**Figure 2 animals-09-00468-f002:**
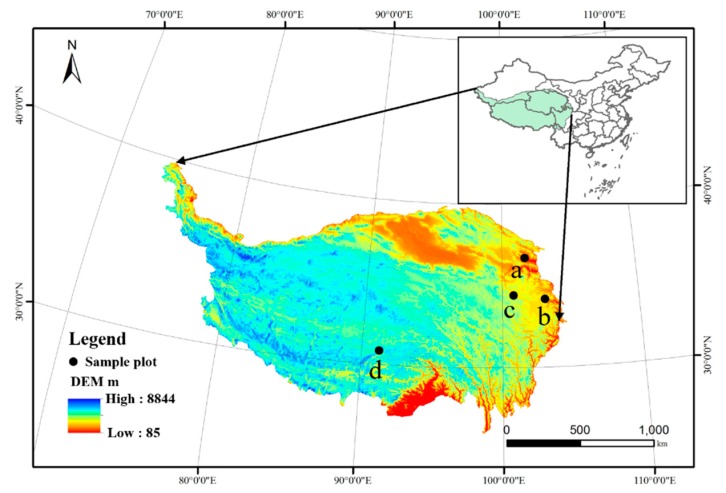
Details of the sampling sites (a: Datong County, b: Hongyuan County, c: Maqu County, and d: Naqu County).

**Figure 3 animals-09-00468-f003:**
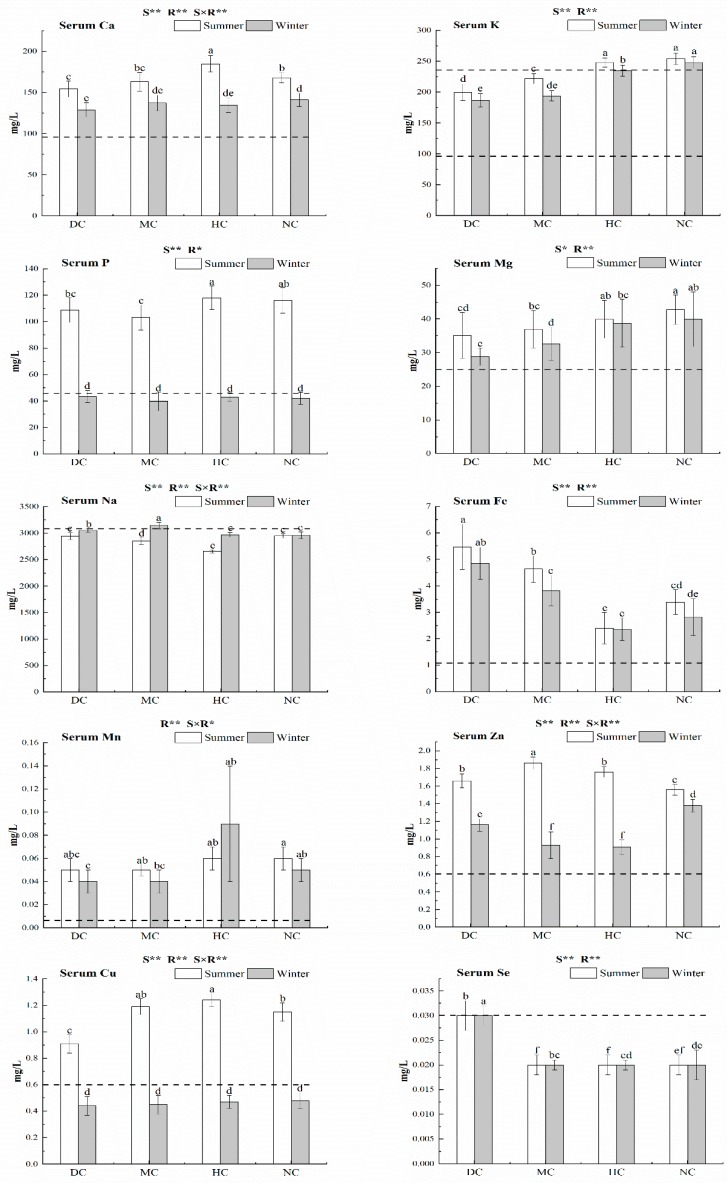
The separate serum Ca, K, P, Mg, Na, Fe, Mn, Zn, Cu, and Se levels of yak required during summer and winter. DC, Datong County; MC, Maqu County; HC, Hongyuan County; NC, Naqu County. R: region; S: season; S × R: season × region; ns: non-significance, *: significance at 0.05 level, **: significance at 0.01 level. One or two dashed horizontal lines across each graph, indicating the marginal ranges. Serum Ca (90 mg/L), Serum K (97.5–234.0 mg/L), Serum P (46.5 mg/L), Serum Mg (25.0 mg/L), Serum Na (3105.0 mg/L), Serum Fe (1.1 mg/L), Serum Mn (0.006 mg/L), Serum Zn (0.6 mg/L), Serum Cu (0.6 mg/L), and Serum Se (0.03 mg/L) [5].

**Table 1 animals-09-00468-t001:** Comparisons of the environmental, ecological, and behavioral data of the animals selected at the experimental sites.

	Datong County	Maqu County	Hongyuan County	Naqu County
Altitude (m)	3100	3504	3900	4700
Annual temperature (℃)	−0.5	1.2	1.1	−1.5
Annual precipitation (mm)	518	620	743	422
Annual sunshine duration (h)	2685	2580	2417	2886
*Dominant plant species of* alpine meadow vegetation	*Elymus nutans* and *Kobresia humilis.*	*Kobresia graminifolia, Elymus nutans, Agrostis species, Poa pratensis, Saussurea species,* and *Anemone species.*	*Saussurea hieracioides* and *Anaphalis lacteal*	*Kobresia pygmaea* and *Elymus nutans*

**Table 2 animals-09-00468-t002:** Available concentrations (mg/kg DM) of mineral elements in soil (mean ± SE).

Region	Season	Ca	K	P	Mg	Na	Fe	Mn	Zn	Cu	Se^2^
Datong County	Summer	369.95 ± 22.46^e^	160.32 ± 16.34^de^	14.48 ± 1.65^a^	13.42 ± 1.39^de^	5.35 ± 1.00^b^	164.74 ± 11.63^de^	12.38 ± 1.03^ab^	7.42 ± 1.16^a^	0.48 ± 0.06^d^	0.15 ± 0.01^b^
	Winter	361.45 ± 32.10^e^	149.32 ± 10.53^e^	14.32 ± 1.09^a^	13.26 ± 2.86^de^	6.59 ± 0.90^a^	154.91 ± 11.45^ef^	10.54 ± 0.88^b^	6.42 ± 0.94^ab^	0.37 ± 0.05^e^	0.16 ± 0.01^a^
Maqu County	Summer	428.58 ± 38.55^c^	177.40 ± 19.09^b^	11.78 ± 1.25^cd^	15.52 ± 1.24^bc^	3.91 ± 0.84^cd^	242.32 ± 11.68^b^	11.86 ± 1.09^ab^	5.37 ± 1.12^bc^	0.63 ± 0.09^c^	0.12 ± 0.01^e^
	Winter	373.70 ± 24.02^e^	163.39 ± 19.94^cd^	13.55 ± 1.31^ab^	12.25 ± 1.60^e^	3.75 ± 0.65^cd^	222.32 ± 9.98^c^	10.72 ± 1.39^b^	4.84 ± 0.92^c^	0.46 ± 0.05^de^	0.10 ± 0.01^f^
Hongyuan County	Summer	392.37 ± 31.84^d^	170.14 ± 18.28^bcd^	12.53 ± 1.12^bc^	14.59 ± 1.20^cd^	4.69 ± 0.75^bc^	178.21 ± 10.03^d^	11.26 ± 4.74^ab^	5.12 ± 0.72^c^	0.72 ± 0.11^bc^	0.13 ± 0.01^cd^
	Winter	419.24 ± 28.73^c^	159.64 ± 19.36^de^	10.64 ± 1.11^d^	17.19 ± 1.12^ab^	5.69 ± 0.92^ab^	148.21 ± 10.37^f^	10.86 ± 1.14^b^	4.67 ± 0.89^c^	0.62 ± 0.10^c^	0.14 ± 0.01^c^
Naqu County	Summer	488.80 ± 35.52^a^	193.93 ± 21.00^a^	8.59 ± 1.14^e^	18.42 ± 1.59^a^	3.35 ± 0.81^d^	283.85 ± 15.84^a^	13.57 ± 1.33^a^	5.83 ± 1.10^bc^	0.85 ± 0.09^a^	0.12 ± 0.01^e^
	Winter	464.63 ± 32.81^b^	173.93 ± 20.52^bc^	12.39 ± 1.14^bc^	17.42 ± 0.99^ab^	4.27 ± 0.85^cd^	233.85 ± 16.08^bc^	12.45 ± 1.07^ab^	4.95 ± 0.89^c^	0.78 ± 0.09^ab^	0.12 ± 0.01^de^
Recommended level ^1^		72.0	37.0	17.0	30.0	-	2.5	5.0	2.5	0.3	0.5
Significance of region or season	S^3^	**	**	*	ns	**	**	ns	*	**	ns
	R^4^	**	**	**	**	**	**	ns	**	**	**
	R × S^5^	**	ns	**	**	ns	**	ns	ns	ns	*

^1^ Recommended level: Recommended level for Ca, K, P, Mg, Na, Fe, Mn, Zn, Cu, and Se from Rojas [19]. ^2^ Se: total concentration for Se in soils. ^3^ S: season; ^4^ R: region; ^5^ R × S: season × region; ns: non-significant, _*_: significant at 0.05 level, _**_: significant at 0.01 level. ^a,b,c,d,e,f^ Means within a column with different superscripts significantly differ (*p* < 0.05).

**Table 3 animals-09-00468-t003:** Macro and micro mineral concentrations (DM) in forage samples (mean ± SE).

Region	Season	Ca	K	P	Mg	Na	Fe	Mn	Zn	Cu	Se
	g kg^−1^					mg kg^−1^					
Datong County	Summer	5.57 ± 1.06 ^c^	14.70 ± 1.09 ^c^	2.71 ± 0.53 ^a^	1.86 ± 0.56 ^abc^	121.47 ± 11.12 ^a^	327.36 ± 16.39 ^c^	69.47 ± 8.29 ^b^	37.70 ± 2.10 ^b^	9.38 ± 0.96 ^c^	0.03 ± 0.01 ^a^
	Winter	6.73 ± 1.07 ^bc^	4.89 ± 0.92 ^d^	0.76 ± 0.10 ^c^	1.56 ± 0.39 ^c^	89.20 ± 12.97 ^bc^	364.86 ± 16.99 ^a^	73.63 ± 13.11 ^ab^	17.70 ± 1.61 ^d^	2.88 ± 0.85 ^e^	0.04 ± 0.01 ^a^
Maqu County	Summer	8.65 ± 1.11 ^a^	15.60 ± 1.10 ^bc^	1.95 ± 0.52 ^b^	1.75 ± 0.62 ^bc^	119.20 ± 19.17 ^a^	296.39 ± 23.01 ^d^	76.58 ± 10.32 ^ab^	41.81 ± 2.03 ^a^	11.26 ± 1.26 ^b^	0.03 ± 0.01 ^b^
	Winter	8.66 ± 1.10 ^a^	3.85 ± 0.66 ^de^	0.67 ± 0.98 ^c^	1.45 ± 0.21 ^c^	86.82 ± 12.31 ^c^	323.39 ± 25.68 ^c^	77.58 ± 13.10 ^ab^	16.64 ± 1.70 ^de^	2.47 ± 0.74 ^e^	0.04 ± 0.01 ^a^
Hongyuan County	Summer	7.86 ± 1.08 ^ab^	16.67 ± 1.15 ^b^	2.20 ± 0.52 ^b^	2.16 ± 0.56 ^ab^	90.94 ± 16.92 ^bc^	286.21 ± 20.53 ^d^	78.45 ± 16.96 ^ab^	39.77 ± 2.08 ^ab^	13.41 ± 0.50 ^a^	0.01 ± 0.00 ^c^
	Winter	8.28 ± 1.11 ^a^	3.63 ± 0.63 ^e^	0.86 ± 0.10 ^c^	1.46 ± 0.16 ^c^	76.44 ± 11.67 ^d^	327.38 ± 31.26 ^c^	80.64 ± 15.13 ^a^	14.61 ± 1.05 ^e^	3.46 ± 0.56 ^e^	0.03 ± 0.01 ^b^
Naqu County	Summer	6.53 ± 1.20 ^bc^	19.68 ± 1.10 ^a^	2.34 ± 0.44 ^ab^	2.34 ± 0.62 ^a^	96.82 ± 18.95 ^b^	316.92 ± 30.19 ^c^	82.64 ± 16.91 ^a^	33.65 ± 2.07 ^c^	11.88 ± 0.82 ^b^	0.03 ± 0.00 ^b^
	Winter	7.63 ± 1.09 ^ab^	4.75 ± 0.57 ^de^	0.64 ± 0.75 ^c^	1.64 ± 0.11 ^bc^	71.81 ± 13.67 ^d^	345.58 ± 38.00 ^b^	81.45 ± 18.93 ^a^	18.32 ± 2.01 ^d^	4.55 ± 0.87 ^d^	0.03 ± 0.01 ^b^
Recommended level ^1^		2.0–11.0	5.0	1.0–3.8	1.3–2.2	800–1200	40	20–25	9–20	4–14	0.04
Significance of region or season	S^2^	*	**	**	**	**	**	ns	**	**	*
	R^3^	**	**	ns	ns	**	**	**	**	**	**
	R × S^4^	ns	**	ns	ns	**	ns	ns	**	**	**

^1^ Recommended level: Recommended mineral nutrient requirements of cattle [20]. When a value is given, the higher values are for rapidly growing, pregnant, or lactating cattle, and the lower values are for those at maintenance or with a low level of production. ^2^ S: season; ^3^ R: region; ^4^ R × S: season × region; ns: non-significant, _*_: significant at 0.05 level, _**_: significant at 0.01 level. ^a,b,c,d,e^ Means within a column with different superscripts significantly differ (*p* < 0.05).

**Table 4 animals-09-00468-t004:** Transfer factor (TF) in plants/soil (mean ± SE).

Region	Season	Ca	K	P	Mg	Na	Fe	Mn	Zn	Cu	Se
Datong County	Summer	15.05 ± 2.81 ^d^	91.78 ± 7.05 ^bc^	190.57 ± 50.71 ^b^	141.74 ± 50.30 ^a^	23.30 ± 4.60 ^bc^	1.99 ± 0.08 ^b^	5.62 ± 0.48	5.22 ± 1.05 ^b^	19.79 ± 3.99 ^a^	0.24 ± 0.30 ^b^
	Winter	18.68 ± 3.27 ^bc^	32.63 ± 4.37 ^d^	53.42 ± 8.67 ^b^	120.87 ± 34.30 ^ab^	13.76 ± 1.96 ^d^	2.37 ± 0.25 ^a^	7.04 ± 0.84	2.82 ± 0.53 ^c^	7.81 ± 2.40 ^c^	0.23 ± 0.05 ^b^
Maqu County	Summer	20.21 ± 2.61 ^b^	87.90 ± 2.89 ^c^	166.82 ± 51.26 ^b^	111.99 ± 34.47 ^ab^	31.64 ± 6.66 ^a^	1.22 ± 0.04 ^d^	6.54 ± 1.24	8.06 ± 1.66 ^a^	18.22 ± 4.30 ^a^	0.22 ± 0.07 ^b^
	Winter	23.72 ± 2.95 ^a^	23.66 ± 4.39 ^e^	49.75 ± 8.44 ^b^	122.27 ± 35.70 ^ab^	23.90 ± 5.07 ^b^	1.46 ± 0.10 ^c^	7.36 ± 1.13	3.50 ± 0.47 ^c^	5.38 ± 1.57 ^c^	0.34 ± 0.06 ^a^
Hongyuan County	Summer	20.04 ± 2.69 ^b^	98.27 ± 9.55 ^ab^	176.95 ± 44.22 ^b^	147.99 ± 35.45 ^a^	19.75 ± 3.15 ^bcd^	1.61 ± 0.14 ^c^	12.73 ± 16.80	7.90 ± 1.26 ^a^	18.95 ± 2.80 ^a^	0.11 ± 0.02 ^c^
	Winter	19.77 ± 2.81 ^b^	22.67 ± 3.21 ^e^	80.92 ± 5.65 ^b^	85.34 ± 14.07 ^b^	13.72 ± 2.22 ^d^	2.21 ± 0.12 ^a^	7.47 ± 0.69	3.21 ± 0.57 ^c^	5.64 ± 1.09 ^c^	0.20 ± 0.05 ^b^
Naqu County	Summer	13.38 ± 2.45 ^b^	101.72 ± 7.67 ^a^	275.12 ± 49.88 ^a^	128.25 ± 35.64 ^ab^	30.30 ± 8.11 ^a^	1.12 ± 0.04 ^d^	6.16 ± 0.94	6.03 ± 1.62 ^b^	14.15 ± 1.97 ^b^	0.24 ± 0.05 ^b^
	Winter	16.40 ± 2.18 ^cd^	27.38 ± 3.28 ^de^	52.32 ± 8.05 ^b^	94.31 ± 8.17 ^b^	17.41 ± 3.64 ^cd^	1.49 ± 0.17 ^c^	6.56 ± 0.69	3.88 ± 0.86 ^c^	5.93 ± 1.42 ^c^	0.18 ± 0.07 ^b^
Significance of region or season	S ^1^	**	**	**	**	**	ns	ns	**	**	*
	R ^2^	**	**	**	ns	**	ns	ns	**	*	**
	R × S ^3^	ns	**	**	ns	ns	ns	ns	**	ns	**

^1^ S: season; ^2^ R: region; ^3^ R × S: season × region; ns: non-significant, _*_: significant at 0.05 level, _**_: significant at 0.01 level. ^a,b,c,d,e^ Means within a column with different superscripts significantly differ (*p* < 0.05).

**Table 5 animals-09-00468-t005:** Transfer factor (TF) in yak/plants (mean ± SE).

Region	Season	Ca	K	P	Mg	Na	Fe	Mn	Zn	Cu	Se
Datong County	Summer	0.03 ± 0.00 ^a^	0.10 ± 0.01 ^d^	0.04 ± 0.01 ^c^	0.02 ± 0.01 ^ab^	24.39 ± 2.16 ^e^	0.02 ± 0.01 ^a^	0.51 ± 0.12	0.04 ± 0.00 ^d^	0.10 ± 0.01 ^c^	0.74 ± 0.15 ^c^
	Winter	0.02 ± 0.00 ^b^	0.04 ± 0.01 ^c^	0.06 ± 0.01 ^ab^	0.02 ± 0.01 ^ab^	34.25 ± 0.96 ^c^	0.01 ± 0.00 ^bc^	0.39 ± 0.04	0.07 ± 0.01 ^b^	0.16 ± 0.06 ^ab^	0.81 ± 0.13 ^bc^
Maqu County	Summer	0.02 ± 0.00 ^b^	0.01 ± 0.00 ^d^	0.06 ± 0.01 ^ab^	0.02 ± 0.01 ^ab^	24.06 ± 2.19 ^e^	0.02 ± 0.01 ^ab^	0.50 ± 0.14	0.04 ± 0.01 ^d^	0.11 ± 0.01 ^c^	0.77 ± 0.19 ^c^
	Winter	0.02 ± 0.01 ^b^	0.05 ± 0.01 ^b^	0.06 ± 0.01 ^ab^	0.02 ± 0.00 ^ab^	36.22 ± 0.91 ^c^	0.01 ± 0.00 ^bc^	0.42 ± 0.07	0.06 ± 0.02 ^bc^	0.21 ± 0.09 ^a^	0.70 ± 0.13 ^c^
Hongyuan County	Summer	0.02 ± 0.00 ^b^	0.02 ± 0.02 ^d^	0.06 ± 0.01 ^ab^	0.02 ± 0.01 ^ab^	29.35 ± 2.22 ^d^	0.01 ± 0.00 ^c^	0.51 ± 0.09	0.04 ± 0.00 ^d^	0.09 ± 0.01 ^c^	1.41 ± 0.32 ^a^
	Winter	0.02 ± 0.01 ^b^	0.07 ± 0.01 ^a^	0.05 ± 0.01 ^bc^	0.03 ± 0.01 ^a^	38.86 ± 0.85 ^b^	0.01 ± 0.00 ^c^	0.48 ± 0.11	0.06 ± 0.01 ^b^	0.14 ± 0.03 ^bc^	0.90 ± 0.21 ^bc^
Naqu County	Summer	0.03 ± 0.01 ^a^	0.01 ± 0.00 ^d^	0.05 ± 0.01 ^bc^	0.02 ± 0.00 ^b^	30.66 ± 2.59 ^d^	0.01 ± 0.00 ^bc^	0.53 ± 0.09	0.05 ± 0.00 ^cd^	0.10 ± 0.01 ^c^	0.74 ± 0.13 ^c^
	Winter	0.02 ± 0.00 ^b^	0.05 ± 0.00 ^b^	0.07 ± 0.01 ^a^	0.02 ± 0.00 ^ab^	41.30 ± 2.24 ^a^	0.01 ± 0.00 ^bc^	0.50 ± 0.11	0.08 ± 0.01 ^a^	0.11 ± 0.02 ^c^	1.07 ± 0.40 ^b^
Significance of region or season	S^1^	**	**	*	ns	**	ns	*	**	**	ns
	R^2^	**	**	ns	ns	**	**	ns	**	*	**
	R × S^3^	ns	**	*	ns	ns	ns	ns	ns	ns	**

^1^ S: season; ^2^ R: region; ^3^ R×S: season × region; ns: non-significant, _*_: significant at 0.05 level, _**_: significant at 0.01 level. ^a,b,c,d,e^ Means within a column with different superscripts significantly differ (*p* < 0.05).

**Table 6 animals-09-00468-t006:** Transfer factor (TF) in yak/soil (mean ± SE).

Region	Season	Ca	K	P	Mg	Na	Fe	Mn	Zn	Cu	Se
Datong County	Summer	0.42 ± 0.03 ^b^	1.25 ± 0.11 ^c^	7.59 ± 1.05 ^c^	2.67 ± 0.61	565.47 ± 100.74 ^cd^	0.03 ± 0.00 ^a^	2.87 ± 0.70	0.23 ± 0.04 ^bc^	1.91 ± 0.25 ^a^	0.17 ± 0.02 ^bc^
	Winter	0.36 ± 0.03 ^cde^	1.25 ± 0.13 ^c^	3.05 ± 0.40 ^de^	2.24 ± 0.37	471.67 ± 72.93 ^d^	0.03 ± 0.00 ^a^	2.75 ± 0.34	0.18 ± 0.03 ^c^	1.22 ± 0.35 ^bc^	0.18 ± 0.02 ^b^
Maqu County	Summer	0.38 ± 0.03 ^c^	1.25 ± 0.07 ^c^	8.78 ± 0.39 ^b^	2.40 ± 0.40	763.96 ± 192.33 ^ab^	0.02 ± 0.00 ^b^	3.11 ± 0.33	0.36 ± 0.07 ^a^	1.92 ± 0.33 ^a^	0.16 ± 0.02 ^bc^
	Winter	0.37 ± 0.04 ^cd^	1.19 ± 0.11 ^c^	2.93 ± 0.34 ^e^	2.71 ± 0.43	863.94 ± 174.17 ^ab^	0.02 ± 0.00 ^b^	3.07 ± 0.50	0.20 ± 0.06 ^c^	1.01 ± 0.27 ^cd^	0.23 ± 0.01 ^a^
Hongyuan County	Summer	0.47 ± 0.03 ^a^	1.46 ± 0.09 ^a^	9.51 ± 1.29 ^b^	2.75 ± 0.28	577.64 ± 88.60 ^cd^	0.01 ± 0.01 ^c^	6.22 ± 7.88	0.35 ± 0.05 ^a^	1.75 ± 0.25 ^a^	0.15 ± 0.02 ^c^
	Winter	0.32 ± 0.02 ^ef^	1.47 ± 0.06 ^a^	4.09 ± 0.64 ^d^	2.26 ± 0.14	532.19 ± 79.99 ^cd^	0.02 ± 0.00 ^b^	3.64 ± 0.92	0.20 ± 0.05 ^c^	0.78 ± 0.19 ^de^	0.17 ± 0.02 ^bc^
Naqu County	Summer	0.34 ± 0.02 ^de^	1.32 ± 0.10 ^bc^	13.65 ± 1.50 ^a^	2.35 ± 0.46	926.41 ± 236.20 ^a^	0.01 ± 0.00 ^c^	3.16 ± 0.31	0.28 ± 0.07 ^b^	1.37 ± 0.20 ^b^	0.17 ± 0.02 ^bc^
	Winter	0.30 ± 0.02 ^f^	1.43 ± 0.12 ^bc^	3.44 ± 0.65 ^de^	2.29 ± 0.13	720.50 ± 166.94 ^cd^	0.01 ± 0.00	3.25 ± 0.82	0.29 ± 0.06 ^ab^	0.63 ± 0.06 ^e^	0.17 ± 0.01 ^bc^
Significance of region or season	S^1^	**	ns	**	ns	ns	ns	ns	**	**	**
	R^2^	**	**	**	ns	**	**	ns	**	**	**
	R × S^3^	**	ns	**	ns	ns	*	ns	**	ns	**

^1^ S: season; ^2^ R: region; ^3^ R × S: season × region; ns: non-significant, _*_: significant at 0.05 level, _**_: significant at 0.01 level. ^a,b,c,d,e,f^ Means within a column with different superscripts significantly differ (*p* < 0.05).

**Table 7 animals-09-00468-t007:** Soil–plant–yak relationship (correlation) with respect to macro and micro mineral status.

	Season		Ca	K	P	Mg	Na	Fe	Mn	Zn	Cu	Se
Plant–yak	Summer	Pearson correlation value	0.493	0.878	0.297	0.830	0.455	0.631	0.926	0.944	0.800	0.526
		*p* value	0.014	0.000	0.327	0.000	0.026	0.001	0.000	0.000	0.000	0.008
	Winter	Pearson correlation value	0.389	0.189	0.356	0.130	0.252	0.610	0.370	0.794	0.621	0.639
		*p* value	0.061	0.376	0.088	0.544	0.236	0.002	0.075	0.000	0.001	0.001
Soil–plant	Summer	Pearson correlation value	0.138	0.873	0.345	0.651	0.331	0.042	0.649	−0.331	0.690	0.404
		*p* value	0.520	0.000	0.099	0.001	0.114	0.845	0.001	0.114	0.000	0.050
	Winter	Pearson correlation value	0.090	0.070	−0.485	0.228	0.286	−0.250	0.698	0.432	0.884	0.062
		*p* value	0.677	0.745	0.016	0.284	0.176	0.239	0.000	0.035	0.000	0.773
Soil–yak	Summer	Pearson correlation value	0.085	0.684	−0.236	0.816	−0.284	−0.197	0.825	−0.442	0.685	0.747
		*p* value	0.693	0.000	0.268	0.000	0.179	0.355	0.000	0.031	0.000	0.000
	Winter	Pearson correlation value	0.821	0.736	0.159	0.966	−0.076	−0.129	−0.034	−0.250	0.879	0.520
		*p* value	0.000	0.000	0.459	0.000	0.725	0.547	0.876	0.239	0.000	0.009

An attempt was made to predict the mineral content in yak whilst keeping the soil and herbage mineral content as independent variables. Prediction equations that could fairly predict the mineral content in yak based on the mineral content in soil and herbage are given in Table 8 and Table 9. Equations developed in the present study for the prediction of K (R^2^ = 0.800), Mg (R^2^ = 0.820), Mn (R^2^ = 0.945), Zn (R^2^ = 0.911), Cu (R^2^ = 0.673), and Se (R^2^ = 0.786) had significant R^2^ values in summer; however, Ca (R^2^ = 0.879), K (R^2^ = 0.748), Mg (R^2^ = 0.942), Mn (R^2^ = 0.945), Zn (R^2^ = 0.642), Cu (R^2^ = 0.873), and Se (R^2^ = 0.800) had significant R^2^ values in winter.

**Table 8 animals-09-00468-t008:** Regression equation of the soil–plant–yak continuum in relation to mineral status in summer.

Mineral	Regression Equation to Predict Mineral Content in Herbage Based on the Mineral Status of Soil	R^2^	Regression Equation to Predict Mineral Content in Yak Based on the Mineral Status of Herbage	R^2^	Regression Equation to Predict Mineral Content in Yak Based on the Mineral Status of Soil	R^2^	Regression Equation to Predict Mineral Content in Yak Based on the Mineral Status of Soil and Herbage	R^2^
Ca	H = 0.004S + 5.618	0.019	Y = 4.670H + 134.132	0.243	Y = 0.021S + 158.585	0.007	Y = 0.004S + 4.648H + 132.473	0.243
K	H = 0.131S−6.317	0.762	Y = 10.649H + 51.001	0.772	Y = 1.248S + 10.095	0.468	Y = −0.634S + 14.330H + 100.625	0.800
P	H = 0.048S + 1.723	0.119	Y = 4.014H + 101.907	0.044	Y = −0.635S + 118.649	0.056	Y = −0.942S + 6.330H + 107.745	0.151
Mg	H = 0.083S + 0.742	0.424	Y = 13.203H + 11.071	0.689	Y = 1.655S + 12.222	0.665	Y = 0.970S + 8.254H + 6.093	0.820
Na	H = 5.782S + 82.097	0.110	Y = 4.523H + 2345.034	0.207	Y = −49.347S + 3042.918	0.081	Y = −84.770S + 6.126H + 2539.962	0.419
Fe	H = 0.014S + 303.606	0.002	Y = 0.046H−10.150	0.398	Y = −0.005S + 5.019	0.039	Y = −0.006S + 0.047H−9.152	0.448
Mn	H = 4.777S + 16.193	0.422	Y = 0.001H−0.047	0.858	Y = 0.009S−0.056	0.681	Y = 0.001S + 0.004H + 0.071	0.945
Zn	H = −1.067S + 44.517	0.110	Y = 0.035H + 0.352	0.892	Y = −0.053S + 2.023	0.195	Y = −0.017S + 0.034H + 0.523	0.911
Cu	H = 7.945S + 6.149	0.477	Y = 0.068H + 0.342	0.640	Y = 0.671S + 0.673	0.649	Y = 0.249S + 0.053H + 0.347	0.673
Se	H = 0.254S−0.007	0.163	Y = 0.076H + 0.002	0.277	Y = 0.197S−0.004	0.558	Y = 0.168S + 0.112H−0.004	0.786

H: mineral content in herbage, S: mineral content in soil, and Y: mineral content in yak.

**Table 9 animals-09-00468-t009:** Regression equation of the soil−plant−yak continuum in relation to mineral status in winter.

Mineral	Regression Equation to Predict Mineral Content in Herbage Based on the Mineral Status of Soil	R^2^	Regression Equation to Predict Mineral Content in Yak Based on the Mineral Status of Herbage	R^2^	Regression Equation to Predict Mineral Content in Yak Based on the Mineral Status of Soil	R^2^	Regression Equation to Predict Mineral Content in Yak Based on the Mineral Status of Soil and Herbage	R^2^
Ca	H = 0.002S + 7.164	0.008	Y = 2.981H + 112.752	0.151	Y = 0.123S + 86.316	0.673	Y = 0.119S + 2.436H + 68.863	0.879
K	H = 0.004S + 3.574	0.005	Y = 9.089H + 176.569	0.036	Y = 2.215S−142.329	0.541	Y = 2.18S + 6.644H−166.071	0.748
P	H = −0.031S + 1.126	0.235	Y = 5.645H + 37.570	0.126	Y = 0.160S + 39.666	0.025	Y = 0.437S + 8.974H + 29.566	0.270
Mg	H = 0.007S + 1.424	0.052	Y = 7.222H + 23.947	0.017	Y = 1.663S + 10.676	0.934	Y = 1.700S−5.247H + 18.145	0.942
Na	H = 1.831S + 71.695	0.082	Y = 6.790H + 2457.451	0.063	Y = −13.113S + 3073.851	0.006	Y = −27.815S + 8.029H + 2498.192	0.295
Fe	H = −0.107S + 360.636	0.063	Y = 0.036H−8.629	0.373	Y = −0.003S + 4.075	0.017	Y = 0.001S + 0.036H−8.869	0.373
Mn	H = 2.802S + 47.111	0.488	Y = 0.004H−0.257	0.137	Y = −0.001S + 0.071	0.001	Y = −0.025S + 0.008H−0.071	0.945
Zn	H = 0.878S + 12.257	0.187	Y = 0.107H−0.701	0.631	Y = 0.068S + 0.739	0.062	Y = −0.031S + 0.113H−0.650	0.642
Cu	H = 4.599S + 0.776	0.782	Y = 0.019H + 0.393	0.396	Y = 0.142S + 0.379	0.773	Y = 0.238S−0.021H + 0.396	0.873
Se	H = 0.021S + 0.027	0.004	Y = 0.294H + 0.015	0.408	Y = 0.080S + 0.014	0.271	Y = 0.074S + 0.280H + 0.006	0.800

H: mineral content in herbage, S: mineral content in soil, and Y: mineral content in yak.

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
