# Peer review of "Mineral Nutritional Status of Yaks (Bos Grunniens) Grazing on the Qinghai-Tibetan Plateau"

_animals, 2019, doi:10.3390/ani9070468_

Round 1
Reviewer 1 Report
The authors conducted a very important study titled "Mineral nutritional status of Yaks (Bos grunniens) grazing in the Qinghai-Tibetan Plateau", which is good for researchers in the field of nutrition and pastoral industry. The whole manuscript is well written and organized. The topic is important and highly related to the journal. It is suggested that this manuscript should be accepted with some minor revisions.
The significance of the study should be addressed more in the introduction part.
As a research study, more information and references about current study should be added to strengthen the introduction part of the manuscript.
The original data should be provided as supporting materials
what's the implementation of grazing industry regarding the results.
Author Response
Dear Reviewer,
Thank you very much for giving us the chance to revise our manuscript. We sincerely appreciate you for the valuable and professional comments that definitely help us to improve our manuscript. We have studied all these comments carefully and made corresponding corrections. We respond these comments point by point as below, and all corrections were shown in the revised manuscript using the track changes function.
Response to Reviewer 1 Comments
The authors conducted a very important study titled "Mineral nutritional status of Yaks (Bos grunniens) grazing in the Qinghai-Tibetan Plateau", which is good for researchers in the field of nutrition and pastoral industry. The whole manuscript is well written and organized. The topic is important and highly related to the journal. It is suggested that this manuscript should be accepted with some minor revisions.
Point 1: The significance of the study should be addressed more in the introduction part.
Response 1: Thanks for your valuable comments and suggestions. We have changed in Line 61-75.
Point 2: As a research study, more information and references about current study should be added to strengthen the introduction part of the manuscript.
Response 2: Thanks for your valuable comments and suggestions. We have changed in Line 61-75.
Point 3: The original data should be provided as supporting materials
Response 3: All data have been tabulated and statically analyzed using standard design already.
Point 4: what's the implementation of grazing industry regarding the results.
Response 4: The results will be used and implemented to yak grazing on the high plateau via the recommendations to the leaders of the commune as well as other stakeholders especially the farmers. Moreover, these results will be used as the baseline for future research related to minerals supplementation.
Sincerely Yours,
Fujiang Hou
Reviewer 2 Report
Comments are reported within the paper. The periods to check are highlighted in Yellow.
Author must pay particular attention to regression means

Author Response
Dear Reviewer,
Thank you very much for giving us the chance to revise our manuscript. We sincerely appreciate you for the valuable and professional comments that definitely help us to improve our manuscript. We have studied all these comments carefully and made corresponding corrections. We respond these comments point by point as below, and all corrections were shown in the revised manuscript using the track changes function.
Response to Reviewer 2 Comments
Point 1: Line 29-33: Delete this period. It is not an Abstract period
Response 1: Thanks for your comments and suggestions. We accept this suggestion.
Point 2: Line 145: what HC means?Response 2: Thanks for your questions. HC means Hongyuan County. We have changed in Line 145.
Point 3: Line 146: SE (standard error) is corrected
Response 3: Thanks for your comments. We have changed in Line 146.
Point 4: Line 149: within a column
Response 4: Done. We have changed in Line 149.
Point 5: Line 156: what NC means?
Response 5: Thanks for your questions. NC means Naqu County. We have changed in Line 156.
Point 6: Line 162: Macro
Response 6: Done. We have changed in Line 162.
Point 7: Line 162: SE (standard error) is corrected
Response 7: Thanks for your comments. We have changed in Line 162.
Point 8: Line 162: ns
Response 8: Done. We have changed in Line 162.
Point 9: Line 166: column
Response 9: Done. We have changed in Line 166.
Point 10: Line 168-174: If you want use the acronyms, the first time you must explain their meaning
Response 10: Thanks for your comments. We have changed in Line 168-174.
Point 11: Line 202, 205, 208: column
Response 11: Thanks for your comments. We have changed in Line 202, 205, 208.
Point 12: Line 218: If you report the P values you don't use the asterisks
Response 12: Thanks for your comments. We have changed in Line 218.
Point 13: Line 225: Why you don't report R square?
Response 13: Thanks for your questions and suggestions. We have changed in Line 225.
Point 14: Line 352: yak serum
Response 14: Done. We have changed in Line 352.
Point 15: Line 356: The regression equation give an idea of how a dependent variable varies according to another independent one. Is it a mistake write "showed positive relationship....". Please correct this sentence. On the another hand, I suggest to authors to utilize these results with much caution because
the only R² is not sufficient to evaluate the goodness of regression.
Response 15: Thanks for your comments and suggestions, we agreed and we will be more cautious in interpretation the data.
Point 16: Line 368: yak serum
Response 16: Done. We have changed in Line 368.
Sincerely Yours,
Fujiang Hou

Reviewer 3 Report
Author studied relationship between mineral nutrient profile of soil, herbage and Yak blood serum. Paper could be interesting for the reader of MDPI journal but I still have some issues with these paper and those can be clarified before paper would accepted.
-Paper needs moderate English language editing.
-Introduction section miss required information on aim and objective of study. Very short information is provided should be added more information.
-Discussion is very long and not concise. Further, many language errors. Suggest shortening the discussion section with highlighting important finding in relation to the previous finding. Please do not repeat results part again in the discussion section.
- Tables should be in a concise format.
Other are added and highlighted directly in the paper

Author Response
Dear Reviewer,
Thank you very much for giving us the chance to revise our manuscript. We sincerely appreciate you for the valuable and professional comments that definitely help us to improve our manuscript. We have studied all these comments carefully and made corresponding corrections. We respond these comments point by point as below, and all corrections were shown in the revised manuscript using the track changes function.
Author studied relationship between mineral nutrient profile of soil, herbage and Yak blood serum. Paper could be interesting for the reader of MDPI journal but I still have some issues with these paper and those can be clarified before paper would accepted.
Point 1: Paper needs moderate English language editing.
Response 1: Thanks for your comments. We have carefully edited the manuscript throughout the manuscript.
Point 2: Introduction section miss required information on aim and objective of study. Very short information is provided should be added more information.
Response 2: Thanks for your valuable comments and suggestions. We have changed in Line 61-75.
Point 3: Discussion is very long and not concise. Further, many language errors. Suggest shortening the discussion section with highlighting important finding in relation to the previous finding. Please do not repeat results part again in the discussion section.
Response 3: Thanks, we have shortened up some parts already.
Point 4: Tables should be in a concise format.
Response 4: Thank you very much for this nice comment. We have changed.
Point 5: Line 40: add "those" after factors
Response 5: Thanks for your comments. We have changed in Line 40.
Point 6: Line 41: replace "in" to "for"
Response 6: Thanks for your comments. We have changed in Line 41.
Point 7: Line 42: replace "the maintenance" to "maintain
Response 7: Thanks for your comments. We have changed in Line 42.
Point 8: Line 45: replace with "well know"
Response 8: Done. We have changed in Line 45.
Point 9: Line 61: The study aims and objectives should be explained in more detail, thus I suggest to add more information here.
Response 9: Thanks for your valuable comments and suggestions. We have changed in Line 61-75.
Point 10: Line 103-107: reference needed
Response 10: Thanks for your comments. We have added the reference.
Point 11: Line 113: reference needed
Response 11: Thanks for your comments. We have added the reference.
Point 12: Line 146: significant letter are difficult to read so suggest to improve font size in table and further make it concise in manner of digits, column length etc
Response 12: Thanks for your comments. We have changed in Line 146.
Point 13: Line 157: replace with "in low range"
Response 13: Thanks for your comments. We have changed in Line 157.
Point 14: Line 156: Most of the herbage samples were deficient in Se all year round?
Response 14: Thanks for your questions. The meaning of this sentence is that most of the herbage Se concentrations lower than the recommended level both summer and winter.
Point 15: Line 162: significant letter are difficult to read so suggest to improve font size in table and further make it concise in manner of digits, column length etc
Response 15: Thanks for your comments. We have changed in Line 162.
Point 16: Line 190-193: Very long and confusing sentence. suggest to rewrite and break down it in short sentences. (The concentration of serum Na was affected significantly by season (P<0.01) cross four counties, however, compared with summer and winter within each county, only in MC and HC counties, the concentration of Na in winter showed a significantly higher than summer (P<0.05).)
Response 16: Thanks for your comments and suggestions, we have changed to the serum Na concentration was variable among counties and was significantly different between in the winter and summer especially in the MC and HC counties.
Point 17: Line 200: significant letter are difficult to read so suggest to improve font size in table and further make it concise in manner of digits, column length etc
Response 17: Thanks for your comments. We have changed in Line 200.
Point 18: Line 203: significant letter are difficult to read so suggest to improve font size in table and further make it concise in manner of digits, column length etc
Response 18: Thanks for your comments. We have changed in Line 203.
Point 19: Line 206: significant letter are difficult to read so suggest to improve font size in table and further make it concise in manner of digits, column length etc
Response 19: Thanks for your comments. We have changed in Line 206.
Sincerely Yours,
Fujiang Hou
Round 2
Reviewer 3 Report
The author has made changes as per my suggestion. I recommend accepting manuscript in the present form.